# Development of novel optical character recognition system to reduce recording time for vital signs and prescriptions: A simulation-based study

**Shoko Soeno**[1], **Keibun Liu**[2]*, **Shiruku Watanabe**[2], **Tomohiro Sonoo**[2], **Tadahiro Goto**[2]

**1** Palliative Care Department, Southern Tohoku General Hospital, Kohriyama, Fukushima, Japan, **2** TXP Medical Co. Ltd., Bunkyo-ku, Tokyo, Japan

* keiliu0406@gmail.com

**Data Availability Statement:** Data relevant to this study are available from Dryad at https://doi.org/10.5061/dryad.fxpnvx10c.

## Abstract

Digital advancements can reduce the burden of recording clinical information. This intra-subject experimental study compared the time and error rates for recording vital signs and prescriptions between an optical character reader (OCR) and manual typing. This study was conducted at three community hospitals and two fire departments in Japan. Thirty-eight volunteers (15 paramedics, 10 nurses, and 13 physicians) participated in the study. We prepared six sample pictures: three ambulance monitors for vital signs (normal, abnormal, and shock) and three pharmacy notebooks that provided prescriptions (two, four, or six medications). The participants recorded the data for each picture using an OCR or by manually typing on a smartphone. The outcomes were recording time and error rate defined as the number of characters with omissions or misrecognitions/misspellings of the total number of characters. Data were analyzed using paired Wilcoxon signed-rank sum and McNemar's tests. The recording times for vital signs were similar between groups (normal state, 21 s [interquartile range (IQR), 17–26 s] for OCR vs. 23 s [IQR, 18–31 s] for manual typing). In contrast, prescription recording was faster with the OCR (e.g., six-medication list, 18 s [IQR, 14–21 s] for OCR vs. 144 s [IQR, 112–187 s] for manual typing). The OCR had fewer errors than manual typing for both vital signs and prescriptions (0/1056 [0%] vs. 14/1056 [1.32%]; p<0.001 and 30/4814 [0.62%] vs. 53/4814 [1.10%], respectively). In conclusion, the developed OCR reduced the recording time for prescriptions but not vital signs. The OCR showed lower error rates than manual typing for both vital signs and prescription data.

## Introduction

Despite the evolution of clinical technologies, traditional methods of information sharing, including paper-based medical records, remain prevalent in many settings. This is particularly evident in prehospital settings, emergency departments, and disaster sites. While the advancement and widespread adoption of mobile devices have enhanced healthcare, particularly in

**Funding:** TXP Medical Co. Ltd. provided support in the form of salaries for SS, SW, TS, and TG. The funder had no role in study design, data collection and analysis, decision to publish, or preparation of the manuscript. The specific roles of these authors are articulated in the 'author contributions' section.

**Competing interests:** The authors have read the journal's policy and have the following competing interests: Keibun Liu was the core research member of TXP Medical Co. Ltd. Tomohiro Sonoo is the Chief Executive Officer of TXP Medical Co. Ltd and reports grants from AI Hospital Research grant from Japan Cabinet Office. Tadahiro Goto is the Chief Scientific Officer of TXP Medical Co., Ltd. Shoko Soeno, Shiruku Watanabe, Tomohiro Sonoo, and Tadahiro Goto received salaries from TXP Medical Co. Ltd. There are no patents, products in development or marketed products associated with this research to declare. This does not alter our adherence to PLOS ONE policies on sharing data and materials.

prehospital settings [1], the handover process from prehospital care to emergency departments often depends on paper documents or manual input to smart device interfaces.

Accurately collecting essential information, including vital signs and prescription lists, as digitized data to assess a patient's medical history is crucial for paramedics and medical staff to improve patient and research outcomes. Furthermore, information must be collected promptly because settings are often in an emergency state, thus requiring frontline paramedics and clinicians to perform multiple tasks while simultaneously gathering information. The current method of paper-based documentation or manual typing with smart devices is time consuming, which results in multiple errors during the information collection and transfer process and subsequently delays definitive treatment. Therefore, the immediate digitization of essential information to capture a scene is urgently needed.

Previous studies have explored the optical character recognition (OCR) technology used to capture vital signs from paper-based encounters or commercial devices such as oximeters and thermometers [2, 3]. A mobile app–based intelligent care system, including OCR, has the potential to reduce health-related issues in patients with chronic kidney disease [4]. However, these studies did not consider the unique conditions of prehospital settings. Such environments present distinct challenges including time constraints and the requirement for emergency medical personnel to use the system to provide treatment. Here we developed a novel OCR system to recognize the characteristics and number of vital signs on ambulance monitors and translate them into digitized data. This system had the ability to recognize the names of medications on a prescription list and link the digitized information with the names in the World Health Organization drug dictionary. The matched names were then stored as digitized data (e.g., anatomical therapeutic chemical classification codes).

This study aimed to compare the newly developed OCR with manual typing using an iPhone in terms of the time and error rate required to record vital signs on an ambulance monitor and medications on a prescription list in an experimental setting. The main targets of this system, including frontline paramedics and emergency department medical staff, were the participants.

## Materials and methods

### Study design and settings

This intra-participant experimental study was conducted from October to December 2021 at the emergency medicine departments of three community hospitals (Hitachi General Hospital, Takasaki Medical Centre, and Southern Tohoku General Hospital) and two fire departments (Takasaki Fire Department, Takasaki, Gunma and Koriyama Fire Department, Koriyama, Fukushima) covering the hospital area. This study was approved by the Institutional Review Board (IRB) of TXP Medical Co., Ltd., which is certified by the Japanese Ministry of Health, Labour and Welfare (IRB no. 21000041; registration no. TXPREC-005).

### Participants

Participants were recruited as volunteers by SS and KL with the help of the supervisory personnel of each department without any exclusion criteria. We collected the participants' characteristics such as age, sex, length of work experience since graduation, emergency medicine board certification status, and type of private smartphone used.

### Application software

We used NEXT Stage ER mobile (NSER mobile; TXP Medical Co., Ltd., Tokyo, Japan), an application aimed at aiding paramedics and emergency physicians share clinical information

between remote devices in prehospital settings [5]. The NSER mobile system was equipped with an OCR tool that recorded vital signs on an ambulance monitor and prescriptions written in a pharmacy notebook. Although not an open-source device, the OCR system is marketed as an integral component of the NEXT Stage ER mobile device by TXP Medical Co., Ltd. Therefore, it is feasible to replicate our findings using a NEXT Stage ER mobile device.

## Study protocol

All participants received 30 min of instructions from the instructor (SS) on how to perform the experiment and used an NSER mobile device as described in the **S1 and S2 Videos**. Although hands-on feedback and additional instructions were provided during the training, no guidance was provided during the actual study. Each participant initially used the OCR system and then manually entered the same information (iPhone 12 or 13; Apple Inc., Cupertino, California). The recording flow is illustrated in **Fig 1**. When using the OCR tool, the participants clicked the New Registration button to start, took a picture of the vital signs monitor or prescription list using the OCR button, and clicked the Finish Recording button. While entering the data manually, the participants clicked the New Registration button, typed in the clinical information, and finished the experiment by clicking the Finish Recording button.

We used six sample pictures: three ambulance monitors showing vital signs (categorized as normal, abnormal, or shock; **S1 Fig**) and three of a pharmacy notebook detailing the prescriptions (with two, four, or six medications; **S2 Fig**). For each picture, the participants recorded the data using OCR or manually into a smartphone. The pictures on the ambulance monitor displayed each patient's blood pressure, pulse rate, oxygen saturation, and respiratory rate. Three sample pictures of the prescription list with two, four, or six medication names were used. The ethics committee waived the requirement for informed consent from the patients or the sources of the sample pictures because the pictures were anonymized. Each participant provided written informed consent before the experiment.

Each user's actions on the system were automatically logged onto a cloud data server specific to this study for calculation of the recording time. The Internet speed at the site was measured during the study whenever possible.

## Outcomes

The primary outcome was the recording time of vital signs or prescription list data. The recording time was defined as the duration that lapsed from the user clicking the New Registration button to clicking the Finish Recording button for the OCR and manual typing processes (**Fig 1**). Time was calculated using data from the automatic logs of the system. The secondary outcome was error rate defined as the ratio of omissions or typographical errors (misrecognitions for OCR or mistakes for manual typing) to the total number of characters. For example, the vital signs comprise five metrics: pulse rate, systolic blood pressure, diastolic blood pressure, partial pressure of oxygen, and respiratory rate. When a participant failed to record a respiratory rate of 18, the number of errors was two. If a participant mistakenly typed a pulse of 82 for a pulse of 72, the number of errors was one.

## Statistical analysis

To calculate the sample size, we conducted a pilot study with two volunteers (senior residents). The results were as follows: for vital signs, the mean time required to record three pictures using OCR was 16.3 s (standard deviation [SD], 2.4 s), while the time required for manual typing was 22.0 s (SD, 4.6 s); for prescriptions, the mean time for recording two pictures (**S2 Fig**; Prescriptions A and B) using OCR was 28.8 s (SD, 9.5 s), while the time required for manual

**Panel A. Recording flow**

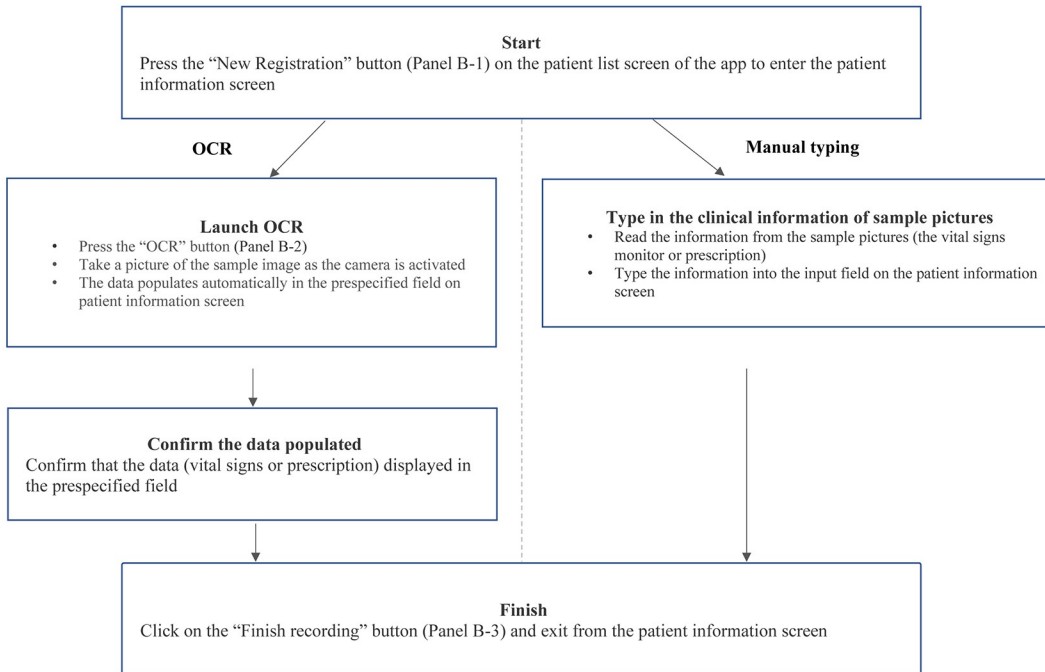

**Panel B. Application Screens**

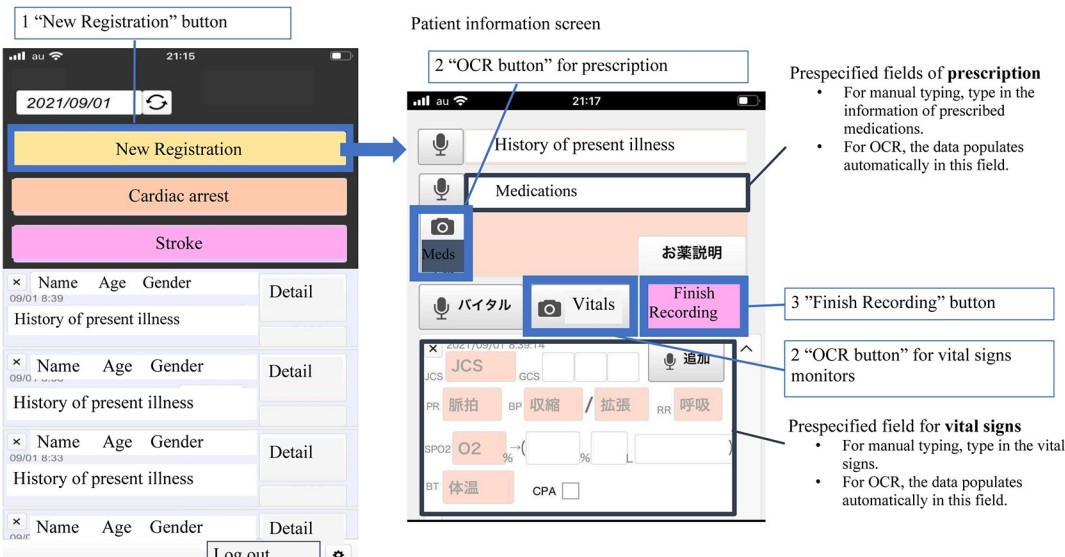

**Fig 1. Recording flow.** A. Recording flow. B. Application screens. **Abbreviations**: OCR, optical character recognition.

typing was 93.5 s (SD, 42.8 s). Based on these numbers, a total of 20 and 14 participants for vital signs and prescription list recording, respectively, would be sufficient to test for differences in time when using a paired t-test with a significance level of 0.05 and power of 0.9. We also conducted a post hoc analysis to ensure that the study was adequately powered. For vital signs, assuming an average input time of 30 s (SD, 10 s) for manual typing and 25 s for OCR, with a significance level ($\alpha$) set at 0.05 and a power ($\beta$) of 0.80, the required sample size was 32

individuals. For prescriptions, we considered a scenario in which OCR reduces input time from an average of 100 s (SD, 30 s) to 50 s, with an α set at 0.05 and a β of 0.80, a sample size of 12 individuals was sufficient.

Categorical variables are expressed as numbers with proportions. Continuous variables are expressed as medians with interquartile ranges (IQR). Intra-person comparisons were used because the recording time of clinical information inherently involves inter-person differences. For the recording time, the Wilcoxon signed-rank test was used to assess the differences in outcomes between the OCR and manual typing groups. For the error rate, McNemar's test was performed to compare the error rates between groups. In cases of missing data, imputation was not applied in the analysis. As we used test-ranked paired observations, comparisons were possible only when the OCR and manual typing values were available. In cases in which either pair was lacking, the data were excluded.

In the sensitivity analyses, we repeated the analysis with stratification by participant age, profession, and Internet speed at the study site.

As a sensitivity analysis, we also performed a complete case analysis in which we excluded all records of participants for whom one or more input values were missing.

The sample size was calculated using R version 4.0.3 (R Foundation, Vienna, Austria). Other analyses were performed using Stata software (version 17.0; StataCorp LP, College Station, TX, USA).

## Results

### Main analysis

Thirty-eight participants (15 paramedics, 10 nurses, and 13 doctors) were recruited. The median age was 34 years (IQR, 27–46 years); of them, 74% (n = 28) were men and 68% used an iPhone daily (**Table 1**). While we assumed 228 paired datasets (38 participants × 6 pictures), nine sets of vital sign monitors and five sets of prescriptions had no records or data errors. Fourteen paired datasets were excluded.

**Table 1. Participants' characteristics.**

| Variable | Overall (N = 38) | Physicians (n = 13) | Nurses (n = 10) | Paramedics (n = 15) |
|---|---|---|---|---|
| Age, years | 34 (27–46) | 32 (29–40) | 26 (24–34) | 37 (36–47) |
| Male sex | 28 (74%) | 10 (77%) | 4 (40%) | 14 (93%) |
| Facility | | | | |
| Hitachi General Hospital | 3 (8%) | 3 (23%) | 0 (0%) | 0 (0%) |
| Southern Tohoku General Hospital | 10 (26%) | 5 (38%) | 5 (50%) | 0 (0%) |
| Takasaki General Medical Center | 16 (42%) | 5 (38%) | 5 (50%) | 0 (0%) |
| Takasaki Fire Department | 0 (0%) | 0 (0%) | 0 (0%) | 6 (40%) |
| Koriyama Fire Department | 0 (0%) | 0 (0%) | 0 (0%) | 9 (60%) |
| Length of experience, years | 9 (5–19) | 6 (4–9) | 5 (2–13) | 15 (12–28) |
| Certified emergency medicine specialist[a] | 20 (53%) | 5 (38%) | 0 (0%) | 15 (100%) |
| Type of private smartphone | | | | |
| iPhone | 26 (68%) | 9 (69%) | 9 (90%) | 8 (53%) |
| Android | 12 (32%) | 4 (31%) | 1 (10%) | 7 (47%) |

[a]Board-certified emergency physician or paramedic

Values are shown as median (interquartile range) or n (%), as appropriate.

**(A) Vital signs**

**(B) prescription**

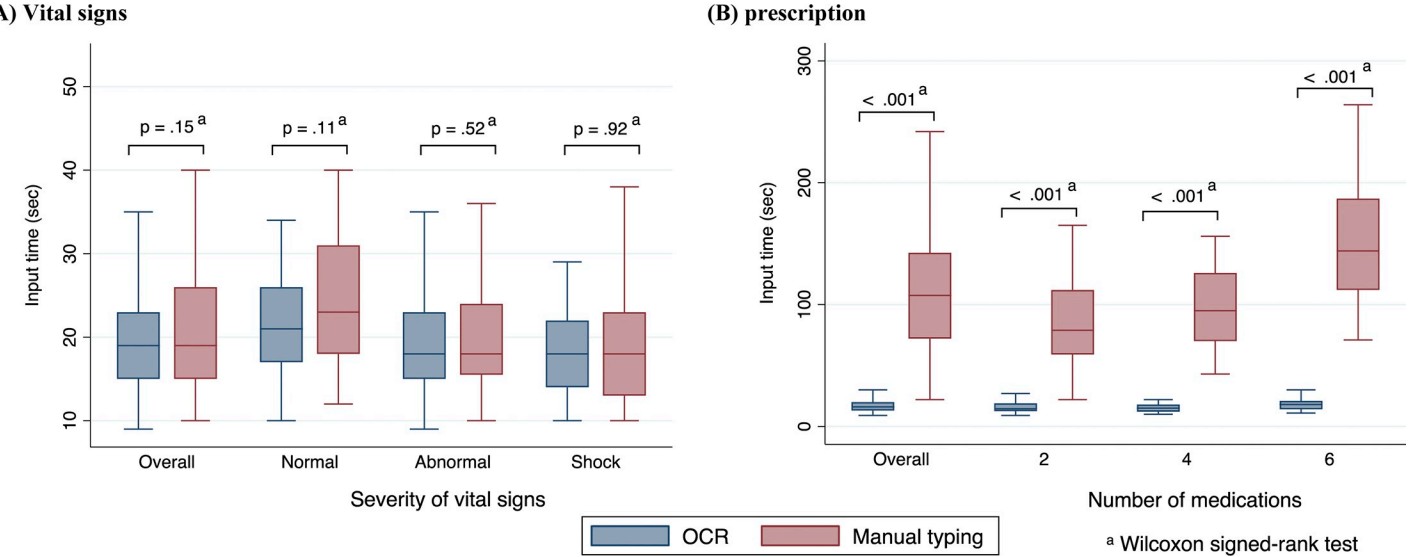

**Fig 2. Recording times of optical character recognition versus manual typing (intra-person comparison, n = 38).** (A) Vital signs and (B) prescriptions. **Abbreviations:** OCR, optical character recognition.

The median recording times for vital signs were similar between groups (normal state, 21 s [IQR, 17–26 s] in the OCR group vs. 23 s [IQR, 18–31 s] in the typing group; p = 0.11; **S1 Table, Fig 2**).

A paired Wilcoxon signed-rank sum test was used to compare the recording times. In contrast, the median recording times for prescriptions were significantly shorter in the OCR group (e.g., six medications list, 18 s [IQR, 14–21 s] in the OCR group vs. 144 s [IQR, 112–187 s] in the manual typing group; p≤0.001; **Fig 2**). In addition, the error rate was significantly lower in the OCR group than in the manual typing group (for a total of three pictures of vital signs, 0/1056 [0%] in the OCR group vs. 14/1056 [1.32%] in the typing group; p<0.001) (**Table 2**). The error rate was lower for the OCR system than for manual typing of recorded prescriptions (30/4814 [0.62%] vs. 53/4814 [1.10%]; p<0.001) (**Table 2**). The characteristics of the omissions differed between groups. If the OCR failed to recognize a drug name, the remaining drug information (amount, shape, and units) was omitted. The OCR tool missed all five medications. During manual typing, participants tended to omit details, such as the shape of the medicine (e.g., tablet or capsule) and units (e.g., mg).

## Age category

The input time by age category showed that participants in their 40s or older completed the input in the same amount of time as other age categories when using OCR, whereas the recording time using manual typing in this group was twice that of participants in their 20s (**Fig 3**).

## Profession

With the OCR, no apparent difference was found in the input time for vital signs or prescriptions among the professionals. When using manual typing, the paramedics tended to require more time to complete the input than the doctors or nurses (**S3 Fig**).

## Internet transmission speed

We measured the Internet transmission speed for 18 participants (14 Mbps for 13 participants and 78 Mbps for five) to investigate the OCR. The input time for vital signs and prescriptions

**Table 2. Error rates.**

| Recording target | OCR group (n = 38) | Manual typing group (n = 38) | P value[a] |
|---|---|---|---|
| Vital signs on the monitor | | | |
| Overall | 0/1056 (0%) | 14/1056 (1.32%) | <0.001 |
| Omissions | 0/1056 (0%) | 12/1056 (1.13%) | |
| Misrecognition/mistyping | 0/1056 (0%) | 2/1056 (0.19%) | |
| Prescription lists | | | |
| Overall | 30/4814 (0.62%) | 53/4814 (1.10%) | <0.001 |
| Omissions | 30/4814 (0.62%) | 46/4814 (1.10%) | |
| Misrecognition/mistyping | 0/4814 (0%) | 7/4814 (0.15%) | |

The error rate is defined as the rate of typographical errors or omissions evaluated by the number of misrecognitions for OCR, misspellings for manual typing, or omissions of the total number of words.

[a]McNemar's test

was significantly shorter when the experiments were performed at a higher Internet transmission speed (p<0.001) (**S2 Table**, **S4 Fig**).

## Sensitivity analysis

A total of 23 participants were included in the sensitivity analysis; only complete cases were used. The results of the sensitivity analysis were consistent with those of the primary analysis of input time (**S3 Table**).

## Discussion

Our findings indicate that using the OCR system notably reduced the prescription list recording time compared with manual typing, with a dose–response relationship; however, there was no difference in the recording time of vital signs irrespective of severity. The error rate of the

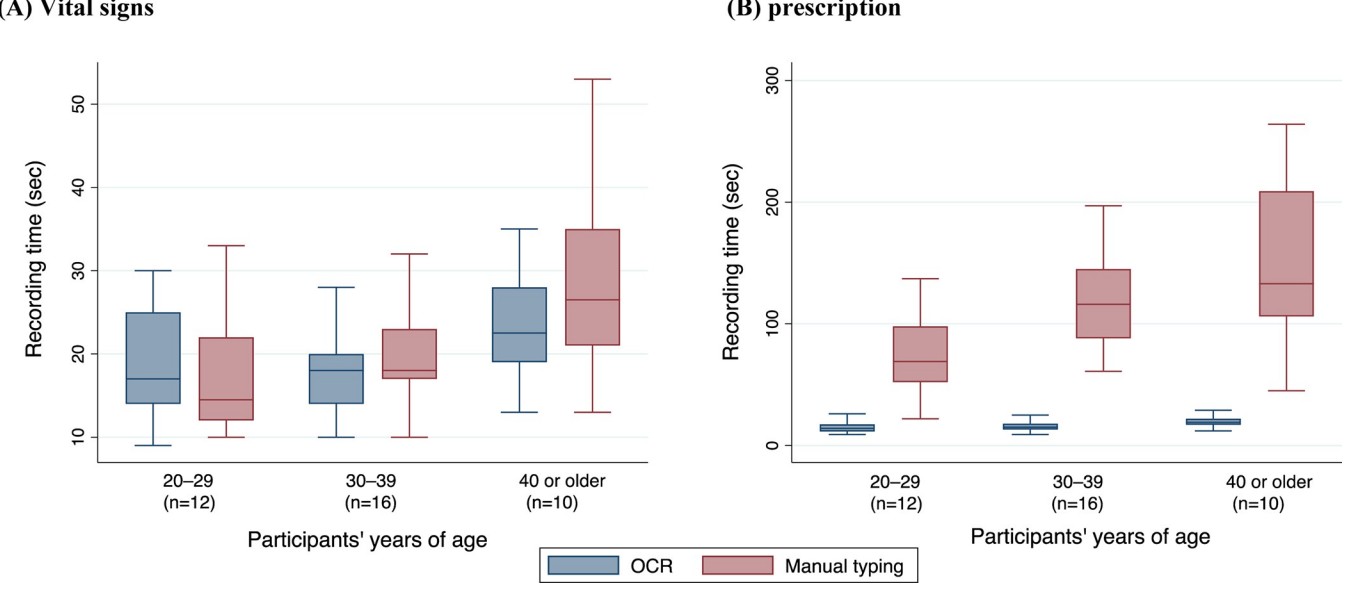

**Fig 3. Recording time by age category.** (A) Vital signs; and (B) Prescription. **Abbreviations:** OCR, optical character recognition.

OCR system was lower than that of manual typing of vital signs and prescriptions. The input time with manual typing was subjective to the participants' ages and professions, whereas the OCR was less affected or unaffected by these variables. Nevertheless, the Internet transmission speed at the study location may have affected the digitalization speed in the OCR system. To the best of our knowledge, this is the first study to examine the application of OCR to record vital signs and prescriptions.

When the OCR system was used to record vital signs on the monitor, the recording time did not change with increasing accuracy. In prehospital settings, vital signs are imperative to deciding whether patients should be transferred and the procedures necessary to stabilize them [6]. Therefore, the timely and accurate digitization and sharing of vital signs are expected to facilitate this process in prehospital and emergency room settings.

The input time of the prescription list was significantly shortened and more accurate with OCR versus manual typing. This beneficial effect is proportional to the number of medicines on the list. Polypharmacy has become a major problem in our aging society; older adults commonly take five or more medicines [7]. Prescriptions provide critical information about a patient's medical history, particularly in cases of unconsciousness [8]. Therefore, the OCR system could aid frontline paramedics and medical staff quickly monitor several medications by researching their drug information.

The error characteristics observed in this study are noteworthy. In the OCR group, the vital signs were not misrecognized, and five medications were omitted. This may indicate that the user must check whether the number of medicines on the list matches the digitized data. The error rate was higher in the manual typing group; however, only drug information details were omitted. Errors that occur with OCR use are often attributed to the photo-resolution [9]. Introducing a system to detect and request the re-uploading of low-resolution photos can help collect training data to mitigate these errors.

Other advantages were observed in the sub-analysis. The OCR system was not affected by user age or profession, which significantly affected the input time for manual typing. These findings indicate that the OCR system can be applied regardless of user characteristics and provide consistent and reliable results. The Internet transmission speed may have influenced the OCR system performance because the captured image of the vital signs or prescription list was sent to the cloud server via the Internet. Further studies involving various types of users and different Internet transmission speeds are necessary.

Digitizing clinical information using an OCR system has several benefits. First, digitization can smooth various procedures, reduce time and cost, and improve patient outcomes. For instance, prehospital transportation time is associated with outcomes among stroke patients [10]. However, less than half of the emergency medical service response times meet United States stroke guidelines [11]. This OCR system outputs structured drug information and instantly assigns codes such as the Anatomical Therapeutic Chemical code [12]. For example, this feature allows paramedics to screen anticoagulants and generic drugs that are contraindications for thrombolytic therapy or endovascular thrombectomy. Second, medical information can be shared and accessed through the cloud. Immediate data sharing facilitates communication between paramedics and clinicians. Third, digitized data can be used in clinical research [13]. If structured data are accumulated in a database, it will become big data for use to answer various clinical questions. Compared with manual data collection, digitization potentially increases data quality and quantity, enabling the performance of novel research. Finally, this OCR technology can be applied to various other clinical settings, such as general screening at scheduled hospital admissions and during a patient's first visit to a clinic without a medical history. OCR/natural language processing (NLP) hybrid usage significantly

improves data extraction efficiency [14, 15]. Thus, the importance of OCR is likely to increase further when used in conjunction with NLP.

Our study has several limitations. First, our dataset featured missing data and outliers with a small sample size, which may have affected the interpretation of the results. However, the exclusion of cases of missing data did not affect its primary outcome (**S3 Table**). A larger sample size is required to validate the reliability of these results. In addition, we did not conduct repeated measurements for each participant, which may have limited the replicability of our findings. Second, only six photographs were used in the experimental setting. The effect and error rate of OCR must be verified in actual prehospital and clinical settings. Third, we used only iPhones to record the data. Fourth, recruiting participants on a volunteer basis may have caused selection bias. And finally, although a pre-calculated sample size was achieved here, the number of participants and occupations remained limited. For instance, medical clerks, who often help document medical information, did not participate in this study. As a next step, we will incorporate this technology into clinical practice, mainly in prehospital or emergency care settings, and investigate its reliability and effectiveness in future studies.

## Conclusions

Using an OCR developed by machine learning with pattern recognition significantly reduced the time required to record prescription lists. Further clinical studies are warranted to confirm the effects of OCR systems on patient outcomes.

## Supporting information

**S1 Video. Explanatory video of the study protocol.**
(MP4)

**S2 Video. Explanatory video of the study protocol.**
(MP4)

**S1 Fig. Sample pictures of vital sign monitoring.** Panel A. Vital signs within the normal state. Panel B. Vital signs in an abnormal state. Panel C. Vital signs in the shock state.
(PDF)

**S2 Fig. Sample pictures of prescriptions in pharmacy notebooks. Prescription A** (**two medications;** total number of characters to count for error rate calculation: 32). **Prescription B** (**four medications;** total number of characters to count for error rate calculation: 55). **Prescription C** (**six medications;** total number of characters to count for error rate calculation: 76).
(PDF)

**S3 Fig. Recording time by profession.**
(PDF)

**S4 Fig. Recording time by Internet transmission speed (megabits per second).**
(PDF)

**S1 Table. Differences in recording time between the optical character recognition and manual typing groups (n = 38).**
(PDF)

**S2 Table. Differences in recording time by Internet transmission speed (megabits per second).**
(PDF)

**S3 Table. Differences in recording time between optical character recognition and manual typing and complete case analysis (n = 23).**
(PDF)

## Acknowledgments

We thank Dr. Hiromu Naraba, Dr. Kensuke Nakamura, and Dr. Hiroshi Machida for recruiting participants at their hospitals. We also thank the paramedics at the Takasaki and Koriyama Fire Departments for participating in this study and Mr. Keita Saegusa for creating the explanatory video.

## Author Contributions

**Conceptualization:** Shoko Soeno, Keibun Liu, Tadahiro Goto.

**Data curation:** Shoko Soeno, Tomohiro Sonoo.

**Formal analysis:** Shoko Soeno, Keibun Liu, Shiruku Watanabe, Tadahiro Goto.

**Funding acquisition:** Tomohiro Sonoo.

**Methodology:** Shoko Soeno, Keibun Liu, Tadahiro Goto.

**Project administration:** Shoko Soeno, Tomohiro Sonoo.

**Resources:** Shoko Soeno.

**Software:** Shoko Soeno, Shiruku Watanabe.

**Supervision:** Tadahiro Goto.

**Visualization:** Shoko Soeno, Keibun Liu, Tadahiro Goto.

**Writing – original draft:** Shoko Soeno, Keibun Liu, Tadahiro Goto.

**Writing – review & editing:** Keibun Liu, Tomohiro Sonoo, Tadahiro Goto.

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
