## [Decision Letter · Decision Letter 0]

27 Jul 2023

PONE-D-22-12113Development of novel optical character recognition system to reduce recording time for vital signs and prescriptions: A simulation-based studyPLOS ONE

Dear Dr. LIU,

Thank you for submitting your manuscript to PLOS ONE. After careful consideration, we feel that it has merit but does not fully meet PLOS ONE’s publication criteria as it currently stands. Therefore, we invite you to submit a revised version of the manuscript that addresses the points raised during the review process.

We look forward to receiving your revised manuscript.

Kind regards,

Asadullah Shaikh, Ph.D.

Academic Editor

PLOS ONE

Journal Requirements:

2. Thank you for providing the following Funding Statement: 

“Dr. Liu reports personal fees from MERA and is the core research member of TXP Medical Co., Ltd completely outside the submitted work. Dr. Sonoo is the Chief Executive Officer of TXP Medical Co. Ltd. and reports grants from AI Hospital Research grant from Japan Cabinet Office. Dr. Goto is the Chief Scientific Officer of TXP Medical Co., Ltd.”

We note that one or more of the authors is affiliated with the funding organization, indicating the funder may have had some role in the design, data collection, analysis or preparation of your manuscript for publication; in other words, the funder played an indirect role through the participation of the co-authors.

If the funding organization did not play a role in the study design, data collection and analysis, decision to publish, or preparation of the manuscript and only provided financial support in the form of authors' salaries and/or research materials, please review your statements relating to the author contributions, and ensure you have specifically and accurately indicated the role(s) that these authors had in your study in the Author Contributions section of the online submission form. Please make any necessary amendments directly within this section of the online submission form.  Please also update your Funding Statement to include the following statement: “The funder provided support in the form of salaries for authors [insert relevant initials], but did not have any additional role in the study design, data collection and analysis, decision to publish, or preparation of the manuscript. The specific roles of these authors are articulated in the ‘author contributions’ section.”

If the funding organization did have an additional role, please state and explain that role within your Funding Statement.

Please also provide an updated Competing Interests Statement declaring this commercial affiliation along with any other relevant declarations relating to employment, consultancy, patents, products in development, or marketed products, etc. 

3. Please amend either the abstract on the online submission form (via Edit Submission) or the abstract in the manuscript so that they are identical

Additional Editor Comments (if provided):

Dear Authors

You are hereby requested to revise the manuscript based on the comments.

Regards

Prof. Dr. Asadullah Shaikh

Reviewers' comments:

Reviewer's Responses to Questions

**Comments to the Author**

1. Is the manuscript technically sound, and do the data support the conclusions?

Reviewer #1: Partly

Reviewer #2: Yes

2. Has the statistical analysis been performed appropriately and rigorously? 

Reviewer #1: Yes

Reviewer #2: Yes

3. Have the authors made all data underlying the findings in their manuscript fully available?

Reviewer #1: Yes

Reviewer #2: No

4. Is the manuscript presented in an intelligible fashion and written in standard English?

Reviewer #1: No

Reviewer #2: Yes

5. Review Comments to the Author

Reviewer #1: 1. The abstract mentioned gives the clear idea about the work. But it should be written in a single paragraph and these headings should be explained later in the manuscript.The abstract in the manuscript should be crisp.

2. Background study of the work should be clearly mentioned in the manuscript. It is expected the authors must add this to the paper.

3. Flow of paper is not well defined.

4. Figure 1 is about recording Flow, it is required to explain the steps in detail in content, it would be better to have a clear understanding of the workflow too.

5. Figure 1 Panel B seems of low quality. Make the figures in higher resolution and the labels can be readable.

6. Give the reference of the data used in Table 1 & 2 if it is taken from some other work or govt. data.

7. Kindly add the authentic reference of the codes of existing models taken in the manuscript.

8. Overall, the manuscript requires major revisions. There are few things to be verified by at author’s end. Like proper citation and referencing wherever the data and facts are mentioned.

9. Similarly, authors are advised to check the grammar as well as spellings. Like in line number 229 on page 12 patent is mentioned instead of patient.

10. Authors should go through journal template also.

Reviewer #2: 1. As mentioned in line no 72 "Using the iphone"; isnt the system applicable on any other smartphone other than iphone.

2.The number of participants taken for sensitivity analysis are very less. How any one can identify the appropriate results with less no of input values provided. Explain

3.Abstract should be more clear for reader's perspective.

4.Authors have provided a comparison with earlier works almost 5 years back, which cannot support this work as per the current scenario.

5.Authors should write the main contributions of the work properly. Authors have written architectural details instead of writing the main contributions.

6.Authors should take care of many typos and grammatical mistakes in the manuscript.

7.A careful proofreading is required to improve the readability of the paper.

8.Authors didn’t include any references from year 2023. An introduction is an important road map for the paper that should be consists of an opening hook to catch the researcher's attention, relevant background study, and a concrete statement that presents main argument but your introduction lacks these fundamentals, especially relevant background studies. This related work is just listed out without comparing the relationship between this paper's model and them; only the method flow is introduced at the end; and the principle of the method is not explained. To make soundness of your study must include these latest related works (years 2022-2023).

9.Mention the limitations and future works of the developed system elaborately.

6. PLOS authors have the option to publish the peer review history of their article (what does this mean?). If published, this will include your full peer review and any attached files.

Reviewer #1: No

Reviewer #2: No

---

## [Decision Letter · Decision Letter 1]

4 Oct 2023

PONE-D-22-12113R1Development of novel optical character recognition system to reduce recording time for vital signs and prescriptions: A simulation-based studyPLOS ONE

Dear Dr. LIU,

Thank you for submitting your manuscript to PLOS ONE. After careful consideration, we feel that it has merit but does not fully meet PLOS ONE’s publication criteria as it currently stands. Therefore, we invite you to submit a revised version of the manuscript that addresses the points raised during the review process.

We look forward to receiving your revised manuscript.

Kind regards,

Asadullah Shaikh, Ph.D.

Academic Editor

PLOS ONE

Journal Requirements:

Reviewers' comments:

Reviewer's Responses to Questions

**Comments to the Author**

1. If the authors have adequately addressed your comments raised in a previous round of review and you feel that this manuscript is now acceptable for publication, you may indicate that here to bypass the “Comments to the Author” section, enter your conflict of interest statement in the “Confidential to Editor” section, and submit your "Accept" recommendation.

Reviewer #1: All comments have been addressed

Reviewer #2: All comments have been addressed

2. Is the manuscript technically sound, and do the data support the conclusions?

Reviewer #1: Partly

Reviewer #2: Yes

3. Has the statistical analysis been performed appropriately and rigorously? 

Reviewer #1: Yes

Reviewer #2: Yes

4. Have the authors made all data underlying the findings in their manuscript fully available?

Reviewer #1: Yes

Reviewer #2: Yes

5. Is the manuscript presented in an intelligible fashion and written in standard English?

Reviewer #1: Yes

Reviewer #2: Yes

6. Review Comments to the Author

Reviewer #1: • The article must present a scientific study that is technically competent and has evidence to back up its findings.

• Rigidly designed experiments with the right controls, replication, and sample sizes must have been used..

• The conclusions must be drawn appropriately based on the data presented.

• Authors have provided a comparison with earlier works almost 5 years back, which cannot support this work as per the current scenario.

• A careful proofreading is required to improve the readability of the paper.

• Latest References not included in the paper.

Reviewer #2: The paper is now acceptable for publication from my side as the author has done the required revision

7. PLOS authors have the option to publish the peer review history of their article (what does this mean?). If published, this will include your full peer review and any attached files.

Reviewer #1: No

Reviewer #2: No

---

## [Author Response · Author response to Decision Letter 1]

16 Nov 2023

The response to reviewers has been attached as Files

---

## [Decision Letter · Decision Letter 2]

11 Dec 2023

Development of novel optical character recognition system to reduce recording time for vital signs and prescriptions: A simulation-based study

PONE-D-22-12113R2

Dear Dr. LIU,

We’re pleased to inform you that your manuscript has been judged scientifically suitable for publication and will be formally accepted for publication once it meets all outstanding technical requirements.

Kind regards,

Asadullah Shaikh, Ph.D.

Academic Editor

PLOS ONE

Additional Editor Comments (optional):

Reviewers' comments:

Reviewer's Responses to Questions

**Comments to the Author**

1. If the authors have adequately addressed your comments raised in a previous round of review and you feel that this manuscript is now acceptable for publication, you may indicate that here to bypass the “Comments to the Author” section, enter your conflict of interest statement in the “Confidential to Editor” section, and submit your "Accept" recommendation.

Reviewer #1: All comments have been addressed

Reviewer #2: All comments have been addressed

2. Is the manuscript technically sound, and do the data support the conclusions?

Reviewer #1: Yes

Reviewer #2: Partly

3. Has the statistical analysis been performed appropriately and rigorously? 

Reviewer #1: Yes

Reviewer #2: Yes

4. Have the authors made all data underlying the findings in their manuscript fully available?

Reviewer #1: Yes

Reviewer #2: Yes

5. Is the manuscript presented in an intelligible fashion and written in standard English?

Reviewer #1: Yes

Reviewer #2: Yes

6. Review Comments to the Author

Reviewer #1: (No Response)

Reviewer #2: The authors have resolved my comments.I am satisfied with the revision made by the authors.good luck

7. PLOS authors have the option to publish the peer review history of their article (what does this mean?). If published, this will include your full peer review and any attached files.

Reviewer #1: **Yes: **Saurabh

Reviewer #2: No

---

## [Editor Report · Acceptance letter]

9 Jan 2024

PONE-D-22-12113R2 

PLOS ONE

Dear Dr. LIU, 

I'm pleased to inform you that your manuscript has been deemed suitable for publication in PLOS ONE. Congratulations! Your manuscript is now being handed over to our production team.

Kind regards, 

on behalf of

Prof. Asadullah Shaikh 

Academic Editor

PLOS ONE